# Malfunctioning CD106-positive, short-term hematopoietic stem cells trigger diabetic neuropathy in mice by cell fusion

Miwako Katagi[1], Tomoya Terashima[1], Natsuko Ohashi[1], Yuki Nakae[1], Akane Yamada[1], Takahiko Nakagawa[1,2], Itsuko Miyazawa[3], Hiroshi Maegawa[3], Junko Okano[4], Yoshihisa Suzuki[4], Kazunori Fujino[5], Yutaka Eguchi[5] & Hideto Kojima [1][✉]

Diabetic neuropathy is an incurable disease. We previously identified a mechanism by which aberrant bone marrow-derived cells (BMDCs) pathologically expressing proinsulin/TNF-α fuse with residential neurons to impair neuronal function. Here, we show that CD106-positive cells represent a significant fraction of short-term hematopoietic stem cells (ST-HSCs) that contribute to the development of diabetic neuropathy in mice. The important role for these cells is supported by the fact that transplantation of either whole HSCs or CD106-positive ST-HSCs from diabetic mice to non-diabetic mice produces diabetic neuronal dysfunction in the recipient mice via cell fusion. Furthermore, we show that transient episodic hyperglycemia produced by glucose injections leads to abnormal fusion of pathological ST-HSCs with residential neurons, reproducing neuropathy in nondiabetic mice. In conclusion, we have identified hyperglycemia-induced aberrant CD106-positive ST-HSCs underlie the development of diabetic neuropathy. Aberrant CD106-positive ST-HSCs constitute a novel therapeutic target for the treatment of diabetic neuropathy.

[1] Department of Stem Cell Biology and Regenerative Medicine, Shiga University of Medical Science, Otsu, Shiga, Japan. [2] Department of Nephrology, Rakuwakai Otowa Hospital, Kyoto, Japan. [3] Department of Internal Medicine, Shiga University of Medical Science, Otsu, Shiga, Japan. [4] Department of Plastic and Reconstructive Surgery, Shiga University of Medical Science, Otsu, Shiga, Japan. [5] Department of Critical and Intensive Care Medicine, Shiga University of Medical Science, Otsu, Shiga, Japan. [✉]email: kojima@belle.shiga-med.ac.jp

Cell-to-cell fusion (cell fusion) is a physiological process which can be observed during organogenesis in utero, as well as after birth[1]. An example would be the process of fertilization, in which a sperm fuses with an egg[2,3]. Postnatally, different types of somatic cells have also been found to fuse with other cells during various physiological processes. Skeletal muscle cells[4,5], osteoclasts[6], syncytiotrophoblasts[7], and Langhans giant cells[8] often appear as multinucleated cells, which are thought to be a product of homotypic cell fusion. Bone marrow-derived cells (BMDCs) can also be a fusion partner to other types of cells, including Purkinje neurons, cardiomyocytes, or hepatocytes[9,10], leading to the formation of heterotypic multinucleated cells.

Cell fusion is often involved in various pathological processes. For example, heterotypic cell fusion has been found to be associated with the development of metastatic cancers[11,12]. In addition, we previously reported that abnormal heterotypic cell fusion occurs between BMDCs and both hepatocytes and neurons, with the latter contributing to the development of diabetic neuropathy (DN)[13,14].

DN, a major complication of diabetes mellitus (DM), is induced at a relatively early stage after the development of diabetes, and progressively worsens throughout life. Patients afflicted with DN often develop gangrene in their lower extremities during the chronic stage, and experience major deterioration in quality of life[15]. A number of pathological mechanisms for the development of DN have been proposed, including ischemia, up-regulation of inflammatory cytokines such as tumor necrosis factor (TNF)-α and interleukin-6 (IL-6), and deficiency of nerve growth factor (NGF), all of which are thought to be involved in the development of DN. To date, therapeutic approaches targeting these pathways have yielded little success. Novel hypotheses need to be tested to enable the formulation of new treatments for DN[16].

In the last dozen years, our group has found that cell fusion between BMDCs and nerve fibers or neurons in the dorsal root ganglion (DRG) may underlie the development of DN in type 1 and type 2 diabetic mouse models[13,17,18]. However, the specific cell type of BMDCs responsible for DN remains elusive. In this study, we categorized bone marrow cells in diabetic mice into specific subtypes and attempted to identify a specific cell type for the development of neuropathy in various diabetic mouse models. We found that CD106-positive cells, a class of hematopoietic stem cell in Lineage (-), c-kit (+) and Sca-1 (+) (LSK) cells, are the culprits that underlie the pathogenesis of DN, and thus constitute a novel therapeutic target for the treatment of this highly prevalent diabetic complication.

## Results

### Fusion of BMDCs with neurons associated with neuronal dysfunction.

Diabetic neuropathy (DN) is characterized by a sensory disorder featuring delayed sensory nerve conduction velocity (SNCV)[19]. In both type 1 and type 2 DM, the mechanism underlying the delay of SNCV has been found to be associated with progressive segmental demyelination and/or axonal degeneration[20]; however, the role of hyperglycemia in the development of nerve dysfunction remains unknown.

We previously reported that abnormal BMDCs are a cause of neuronal injury in diabetes, based on our data that BMDCs exposed to diabetic conditions migrate to the DRG and fuse with DRG neurons[13]. Interestingly, the BMDCs in mice with diabetic neuropathy were found to express proinsulin[21] and TNF-α[13], and caused chronic inflammation at the fusion site, leading to neuronal dysfunction[22–25]. However, the overlap phenomenon could also be accounted for by an alternative possibility that BMDCs could have directly differentiated into neurons in the absence of cell fusion.

In order to address this issue, we created a mouse model in which a marker (red fluorescence) would be created only by the cell fusion process. We therefore transplanted total bone marrow (TBM) from Ayu-1 Cre transgenic mice into LoxP-stop-LoxP-tdTomato mice, which were transgenic mice with the *tdTomato* gene inserted downstream of the *STOP* sequence in the LoxP site (Ayu-1 TBM BMT to lsl-tomato) (Fig. 1a). If cell fusion had occurred in this mouse model, the Cre-recombinase driven by the Ayu-1 promoter[26] would react with LoxP, removing the floxed stop sequence, enabling the expression of tdTomato. After inducing diabetes in these mice using STZ (confirmed by hyperglycemia and delayed SNCV 12 weeks later, Fig. 1b, c), we examined deletion of the stop sequence by PCR. As shown in Fig. 1d, a 545 bp PCR product was detected in DRG neurons of nonDM mice, but not in those of STZ-DM mice, suggesting that the stop sequence in the nuclei of DRG neurons had been deleted by the Cre-recombinase of BMDCs as a consequence of cell fusion between these cells. Consistent with this scenario, histological analyses also revealed the occurrence of tdTomato fluorescence (red) in the DRG neurons of STZ-DM recipients but not in nonDM recipients (Fig. 1e), indicating cell–cell fusion between the donor and recipient cells (Fig. 1e, f).

Previously, we demonstrated that BMDCs expressing proinsulin/TNF-α were also positive for CD45[13]. This time, we confirmed that this type of cell was detected in both type of DM mice, but not in nonDM mice, and in particular in the hematopoietic stem cell fraction (Lineage (-), c-kit (+) and Sca-1 (+): LSK) (Supplementary Fig. 1)[18]. These results suggested that the presence of type 1 and type 2 DM is associated with the appearance of an abnormal cell fraction among the hematopoietic stem cells in the bone marrow.

### Diabetic bone marrow cells can cause neuropathy in non-diabetic mice.

We reasoned that functionally-impaired BMDCs in neural tissues are likely derived from abnormal hematopoietic stem cells in the bone marrow, which would be a key factor for the incurableness of DN only by glycemic control. In order to further investigate the pathological characters of BMDCs, we next isolated TBM cells from STZ-DM mice, transplanted them into wild type mice, and investigated whether diabetic BMDCs would lead to cell fusion and induce neuropathy even in normoglycemic mice (Supplementary Fig. 2).

First, we compared nonDM mice transplanted with TBM cells of STZ-DM mice (TBM-T from STZ-DM) with nonDM mice transplanted with total bone marrow cells of nonDM mice (TBM-T from nonDM) (Supplementary Fig. 2a). Interestingly, TBM-T from STZ-DM donors reduced SNCV by 29% compared with TBM-T from nonDM donors, although the blood glucose concentrations of these two groups of mice were very similar and well within the normal range (Supplementary Fig. 2b, c). We found some GFP-positive cells (green), which were derived from bone marrow, around DRG neurons in both TBM-T from nonDM and TBM-T from STZ-DM (Supplementary Fig. 2d). In contrast, proinsulin-positive (red) and TNF-α-positive (red) cells were observed in DRG neurons only in those receiving TBM-T from STZ-DM mice (Supplementary Fig. 2d middle and bottom panels). While MAP2 expression (red) was observed in many cells of DRG neurons in both types of mice, overlap of GFP signals with MAP2 was seen only in donors receiving TBM-T from STZ-DM mice (Supplementary Fig. 2d top panels). Furthermore, proinsulin- (red) and TNF-α-positive (red) staining were also co-localized in GFP-positive (green) cells (Supplementary Fig. 2d middle and bottom panels). These results indicate that BMDCs from STZ-DM mice had fused with DRG neurons and the fusion cells expressed proinsulin and TNF-α. In other words, bone

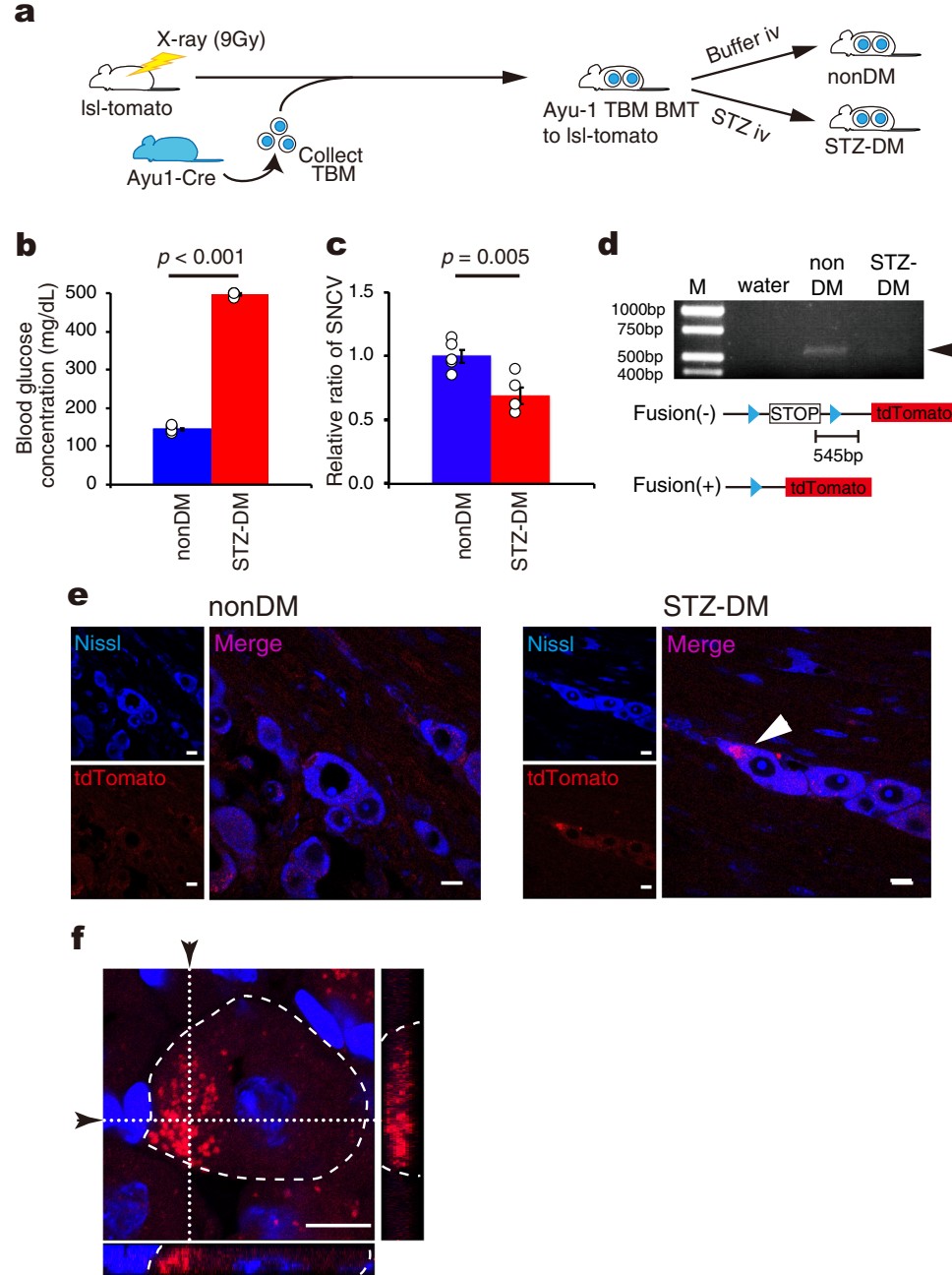

**Fig. 1 The detection of Cell fusion in the new bone marrow transplantation model with streptozotocin-induced diabetic mice (Type 1 diabetic model). a** A schematic experimental design for a mouse model to detect cell fusion. TBM cells (4 × 10⁶ cells) obtained from Ayu1-cre transgenic (Ayu1-cre) mice were transplanted into 9 Gy lethally-irradiated normoglycemic Lox-stop-Lox tdTomato (lsl-tomato) mice (Ayu-1 TBM BMT to lsl-tomato). Four weeks after BMT, Type1 diabetes mellitus was induced by intravenous STZ injection (STZ-DM) while intravenous citrate buffer injection was used as a control (nonDM). **b** Blood glucose concentration of nonDM (*n* = 5) and STZ-DM mice (*n* = 4). **c** Relative ratio of sensory nerve conduction velocity (SNCV) in the sciatic nerve of STZ-DM (*n* = 4) and nonDM (*n* = 5). **d** PCR analysis of genomic DNA from DRG neurons. While nonDM showed the presence of a 545 bp PCR product, STZ-DM did not (Arrow). Schematic illustrating the association between the Cre-loxP system, primers, and PCR products. Blue arrowheads showed loxP sites. (We provide the full gel image of electrophoresis as Supplementary Fig. 5.) **e** Histological analysis of DRG in both nonDM and STZ-DM. Blue showed Nissl stain. Red showed tdTomato. Arrowhead indicates a tdTomato-positive DRG neuron. Scale bars = 10 μm. **f** Z-stack imaging of DRG from STZ-DM. Narrow images indicate the cross-section of *x*-axis or *y*-axis (whited dotted line) in DRG neuron surrounded by white dashed line. Arrows indicate the position of cross-section. Red showed tdTomato. Blue showed nucleus. Scale bar = 10 μm. Data are indicated as means ± SE.

marrow cells that had been exposed to hyperglycemic conditions in donor mice directly contributed to cell fusion and neuronal dysfunction (and neuropathy) in nondiabetic recipient mice which never exhibited hyperglycemia throughout the experiment. TBM cells transiently exposed to diabetic conditions were capable of inducing DN even in nonDM mice. These abnormal cells

migrated to peripheral nerves via the systemic circulation to fuse with neurons in a recipient mouse with normoglycemia.

Since we found that LSK cells in diabetic mice included abnormal cells expressing proinsulin and TNF-α (Supplementary Fig. 1c, d, g, h), we next tested the potent effects of LSK cells of diabetic mice using two different models of nonDM mice; one

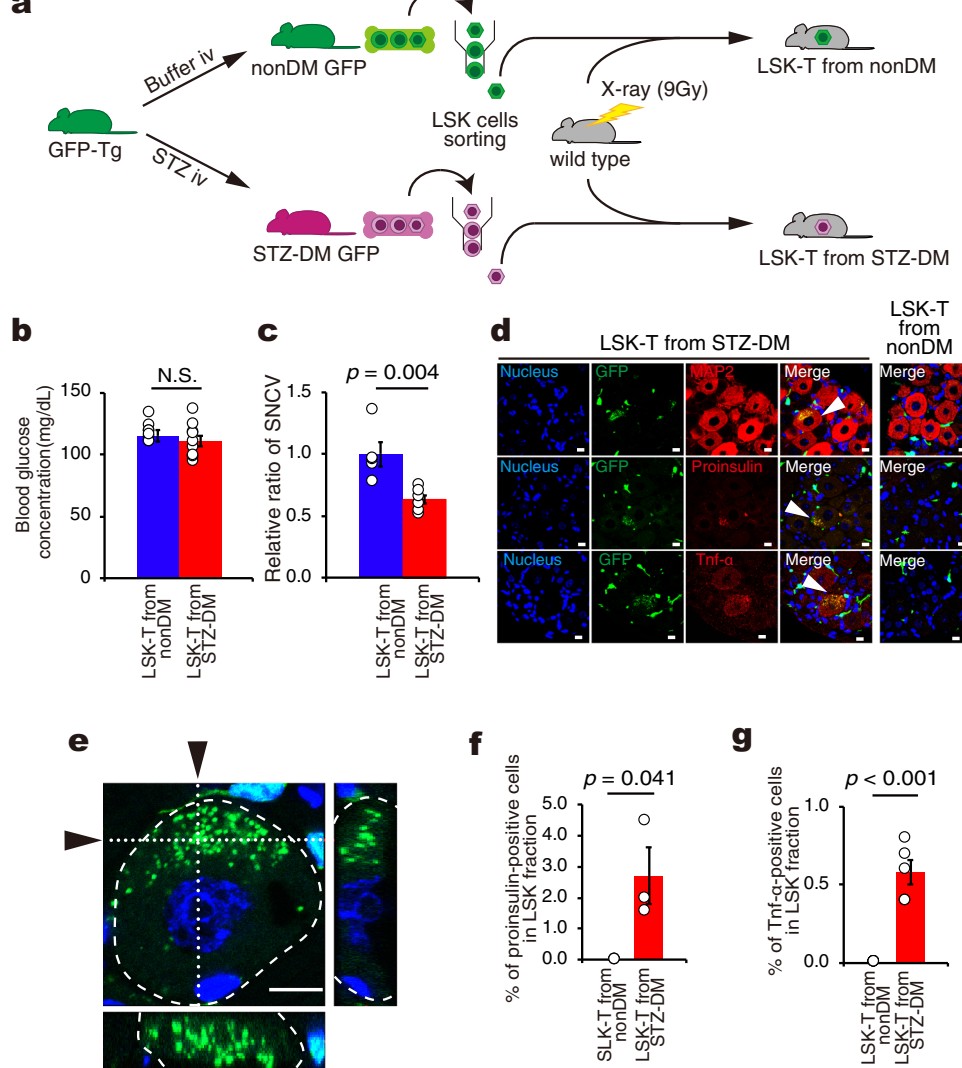

**Fig. 2 Cell fusion and development of neuropathy in mice transplanted with LSK cells from diabetic mice. a** A schematic experimental design for the transplantation of nonDM/STZ-DM LSK cells into normoglycemic mice. Diabetes (DM) was induced by STZ in GFP-Tg mice (STZ-DM GFP) while intravenous citrate buffer injection was used for the nonDM controls (nonDM GFP). Three months later, LSK cells obtained either from nonDM or STZ-DM mice were transplanted into 9 Gy lethally-irradiated normoglycemic wild type mice (LSK-T from nonDM or LSK-T from STZ-DM, respectively). **b** Blood glucose concentration of LSK-T from nonDM ($n = 8$) and LSK-T from STZ-DM ($n = 10$). N.S means not significant. **c** Relative ratio of SNCV in the sciatic nerve following LSK-T from STZ-DM mice ($n = 6$) compared to LSK-T from nonDM donors ($n = 5$). **d** Immunofluorescent staining of the dorsal root ganglion (DRG) showing nuclei (blue), GFP (green), MAP2 (red), proinsulin (red), and TNF-α (red), and merged images in LSK-T from nonDM and LSK-T from STZ-DM mice. Arrowheads showed fusion cells. Scale bars = 10 μm. **e** Z-stack imaging of DRG from LSK-T from STZ-DM. Narrow images indicate the cross-section of x-axis or y-axis (whited dotted line) in DRG neuron surrounded by white dashed line. Arrows indicate the position of cross-section. Green showed GFP. Blue showed nucleus. Scale bar = 10 μm. **f** FACS analysis of proinsulin-positive cells in the LSK fraction from mononuclear cells in LSK-T from nonDM and LSK-T from STZ-DM. The percentage of proinsulin-positive cells among LSK cells ($n = 3$ per group). **g** FACS analysis of TNF-α-positive cells among LSK cells from mononuclear cell in LSK-T from nonDM and LSK-T from STZ-DM. The percentage of TNF-α-positive cells among LSK cells ($n = 5$ per group). Data are indicated as means ± SE.

was nonDM mice with transplantation of LSK cells from STZ-DM mice (LSK-T from STZ-DM) and the other was nonDM mice with transplantation of LSK cells from nonDM mice (LSK-T from nonDM) (Fig. 2a). NonDM mice with transplantation of LSK cells from STZ-DM donors exhibited a decrease in SNCV of 36% in comparison with transplantation of LSK cells from nonDM although blood glucose concentration was identical and within the normal range in the two types of mice (Fig. 2b, c). LSK-T from nonDM as well as LSK-T from STZ-DM resulted in GFP-positive cells (green) around the DRG neurons (Fig. 2d); however, cells with proinsulin (red) and/or TNF-α expression

(red) were present only at the DRG neurons in LSK-T from STZ-DM mice (Fig. 2d middle and bottom panels). Immunostaining for MAP2 (red) demonstrated that LSK-T from STZ-DM, but not LSK-T from nonDM mice, showed yellow signals, indicating the cell fusion of MAP2-positive neurons with GFP-positive BMDCs (green) (Fig. 2d top panels). Furthermore, both proinsulin- (red) and TNF-α-positive (red) reactions were simultaneously co-localized in GFP-positive (green) cells (Fig. 2d middle and bottom panels) so that fused cells aberrantly expressed the two factors. Given the fact that mice with LSK-T from STZ-DM donors exhibited cell fusion between DRG neurons and BMDCs, it is

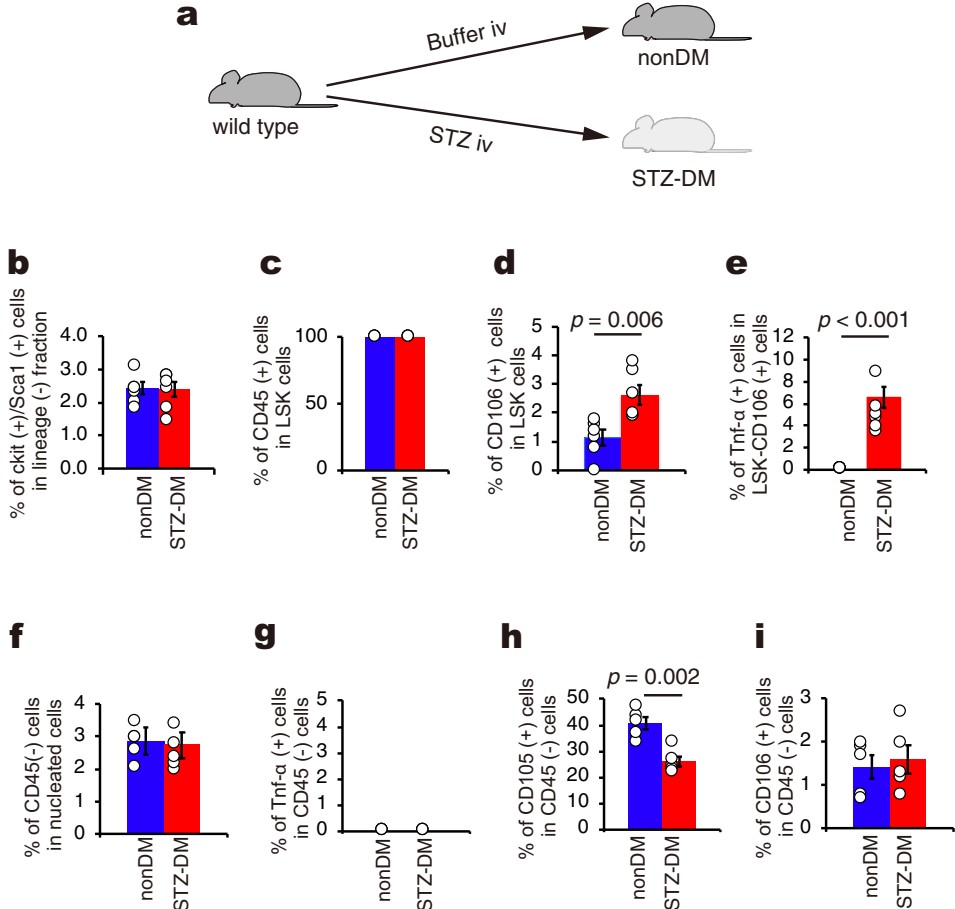

**Fig. 3 Localization of CD45-, CD106-, and TNF-α-positive cells in the bone marrow stem cell fraction. a** A schematic experimental design for the generation of diabetic mice (STZ-DM) and nonDM mice. **b** FACS analysis of LSK cell in nonDM and STZ-DM. Percentage of LSK cells in the mononuclear cell fraction from total bone marrow in nonDM and STZ-DM. ($n = 6$ per group). **c** FACS analysis of CD45-positive cell in LSK cells from nonDM and STZ-DM. Percentage of CD45-positive cells in the LSK cell fraction from mononuclear cells in nonDM and STZ-DM ($n = 3$ per group). **d** FACS analysis of CD106-positive cell in LSK cells from nonDM and STZ-DM. Percentage of CD106-positive cells in the LSK cell fraction from mononuclear cells in nonDM and STZ-DM ($n = 6$ per group). **e** FACS analysis of TNF-α-positive cells in LSK-CD106 cells from nonDM and STZ-DM. Percentage of TNF-α-positive cells in the LSK-CD106 cell fraction from mononuclear cells in nonDM and STZ-DM ($n = 6$ per group). **f** FACS analysis of CD45-negative cell in nucleated cells in nonDM and STZ-DM total bone marrow. Percentage of CD45-negative cells in the nucleated cell fraction in nonDM and STZ-DM total bone marrow ($n = 5$ per group). **g** FACS analysis of TNF-α-positive cells in CD45-negative cells from nonDM and STZ-DM. Percentage of TNF-α-positive cells in the CD45-negative cell fraction from total bone marrow in nonDM and STZ-DM ($n = 3$ per group). No TNF-α-positive cells were detected any mice. **h** Percentage of CD105-positive cells in the CD45-negative cell fraction from total bone marrow in nonDM and STZ-DM ($n = 5$ per group). **i** Percentage of CD106-positive cells in the CD45-negative cell fraction from total bone marrow in nonDM and STZ-DM ($n = 5$ per group). Data are indicated as means ± SE.

likely that diabetic neuropathy resulted from cell fusion between the abnormal BMDCs and the local neurons (Fig. 2d, e).

Next, we examined the phenotype of LSK cells inside the bone marrow in these mice. In mice with LSK-T from STZ-DM donors, 2.7% and 0.6% of LSK cells were positive for proinsulin and TNF-α expression, respectively. In contrast, both markers were negative in LSK cells of mice following LSK-T from nonDM donors (Fig. 2f, g).

**Abnormal hematopoietic stem cells (HSCs) in the CD106 fraction**. In the next set of experiments, we further characterized LSK cells from STZ-DM mice (Fig. 3a). We found that abnormal bone marrow cells were all positive for CD45 (Fig. 3b, c), indicating that these cells belong to the hematopoietic stem cell (HSC) population. In order to identify any factors associated with cell fusion, the gene expression profile of LSK cells was analyzed using microarray technology in which gene ontology (GO) using

the key term "fusion" (Supplementary Fig. 3, Gene Expression Omnibus (GEO) accession number: GSE117088) was applied. Using this analysis, we identified *Vcam1* (*CD106*), *Caveolin2*, *DC-STAMP* and *Neogenin*, all of which were highly expressed in the LSK cells of STZ-DM mice compared with nonDM animals. Interestingly, these factors have been shown to be involved in cell fusion between progenitor cells and their maturation[27–30]. Among these factors, we decided to focus on CD106 since this molecule is a marker of activated endothelial cells and activation of those cells is involved in the pathological process in diabetes[31,32] and vascular formation[33]. Previous studies also indicate the importance of adhesion molecule in hematopoietic stem/progenitor cells (HSPC)[34,35]. Fluorescent activated cell sorting (FACS) analysis of TBM revealed that the number of CD106-positive LSK cells from STZ-DM mice was two times higher than that from nonDM mice (Fig. 3d). Interestingly, CD106-positive LSK cells were also found to express TNF-α from STZ-DM mice (Fig. 3e).

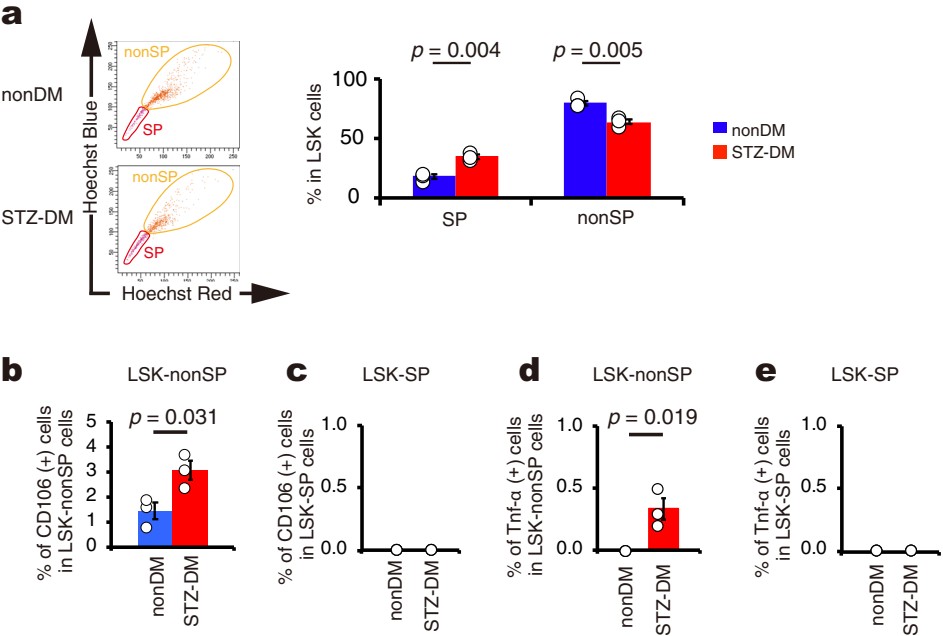

**Fig. 4 CD106- or TNF-α-positive cells exist in the nonSP fraction of LSK cells in STZ-DM. a** FACS analysis of SP and/or nonSP fraction in the LSK cells from nonDM and STZ-DM (left panel, SP cells surrounded by red circle, nonSP cells surrounded by orange circle). Percentage of SP fraction or nonSP fraction in LSK cells from mononuclear cells in nonDM and STZ-DM (right panel, $n = 3$ per group). **b** FACS analysis of CD106-positive cells in LSK-nonSP cells from nonDM and STZ-DM. Percentage of CD106-positive cells in LSK-nonSP cells from mononuclear cells in nonDM and STZ-DM ($n = 3$ per group). **c** FACS analysis of CD106-positive cells in LSK-SP cells from nonDM and STZ-DM. Percentage of CD106-positive cells in LSK-SP cells from mononuclear cells in nonDM and STZ-DM ($n = 3$ per group). No CD106-positive cells were detected in any of the mice. **d** FACS analysis of TNF-α-positive cells in LSK-nonSP cells from nonDM and STZ-DM. Percentage of TNF-α-positive cells in LSK-nonSP from mononuclear cells in nonDM and STZ-DM ($n = 3$ per group). **e** FACS analysis of TNF-α-positive cells in LSK-SP cells from nonDM and STZ-DM. Percentage of TNF-α-positive cells in LSK-SP from mononuclear cells in nonDM and STZ-DM ($n = 3$ per group). No TNF-α-positive cells were detected in any mice. Data are indicated as means ± SE.

In terms of the CD45-negative fraction (a subset of mesenchymal stem cells), all cells were negative for TNF-α in both STZ-DM and nonDM mice (Fig. 3f, g). Among CD45-negative cells, the number of CD105-positive cells, which can be a major component of the vascular niche and a precursor of pericytes[36], was significantly reduced under diabetic conditions (Fig. 3h). In contrast, the number of CD106-positive cells in the CD45-negative fraction was very low in both types of mice and there was no difference in the number of CD106-positive cells between nonDM and STZ-DM mice (Fig. 3i). Our data are consistent with those in previous reports showing that a decrease in CD105-positive cells in the diabetic CD45-negative fraction was observed in blood vessels in diabetic retinopathy to account for pericyte loss[37,38].

**Abnormal short-term hematopoietic stem cells appear in diabetes.** In general, HSCs can be divided into two fractions; long-term HSCs (LT-HSCs) and short-term HSCs (ST-HSCs)[39,40]. LT-HSCs mostly exist in the side population (SP) of LSK cells whereas ST-HSCs exist in the remaining fraction (nonSP)[41]. We first examined the population of SP and nonSP in LSK cells from nonDM and STZ-DM. STZ-DM mice showed an increase in the SP fraction and a decrease in the nonSP fraction compared with nonDM mice (Fig. 4a). We further found that CD106-positive cells were present only in the nonSP fraction in both STZ-DM and nonDM mice; however, the percentage of CD106-positive cells in the nonSP fraction was significantly increased under diabetic conditions compared with nonDM conditions (Fig. 4b, c). With respect to TNF-α expression, cells in the LSK-SP fraction were negative in both STZ-DM and nonDM mice, while some cells in the nonSP fraction of STZ-DM, but not nonDM mice, were positive for TNF-α (Fig. 4d, e). Taken together, these data

show that target cells with TNF-α expression are present only in the CD106-positive fraction of ST-HSCs in diabetic mice.

**Diabetic CD106-positive HSCs can produce neuropathy in nondiabetes.** Since the number of CD106-positive cells increased in the CD45-positive fraction of LSK cells from STZ-DM mice (Fig. 3d), we postulated that this cell type could be the true culprit that underlies the development of DN, and therefore designed an experiment to address this issue. Diabetes was induced by STZ in GFP-Tg mice, and CD106-positive and CD106-negative cells were isolated from the LSK cell fraction of diabetic mice. Then, the two types of cell were transplanted separately into nonDM mice (Fig. 5a). Again, all recipient mice remained normoglycemic throughout the entire experiment (Fig. 5b). The transplantation of DM-CD106-positive cells decreased SNCV by 16% in nonDM mice compared with that of DM-CD106-negative cells (Fig. 5c). Similar to earlier experiments in this study, GFP-positive BMDCs (green) were detected around DRG neurons of both nonDM with DM-CD106-negative cells and nonDM with DM-CD106-positive cells (Fig. 5d); however, the expression of either proinsulin (red) or TNF-α (red) cells at DRG neurons was detected only in nonDM with DM-CD106-positive cells (Fig. 5d. middle and bottom panels). While all neurons were positive only for MAP2 in nonDM with DM-CD106-negative cells, some neurons showed a yellow color in nonDM with DM-CD106-positive cells, suggesting that GFP-positive BMDCs had merged with MAP2-positive neurons (Fig. 5d top panels). Furthermore, either proinsulin (red) or TNF-α (red) was also expressed in GFP cells (Fig. 5d middle and bottom panels). Moreover GFP-positive signal was expressed inside DRG neuron (Fig. 5e). These results indicated that neuropathy was caused by the abnormal cells derived from diabetic CD106-positive abnormal stem cells.

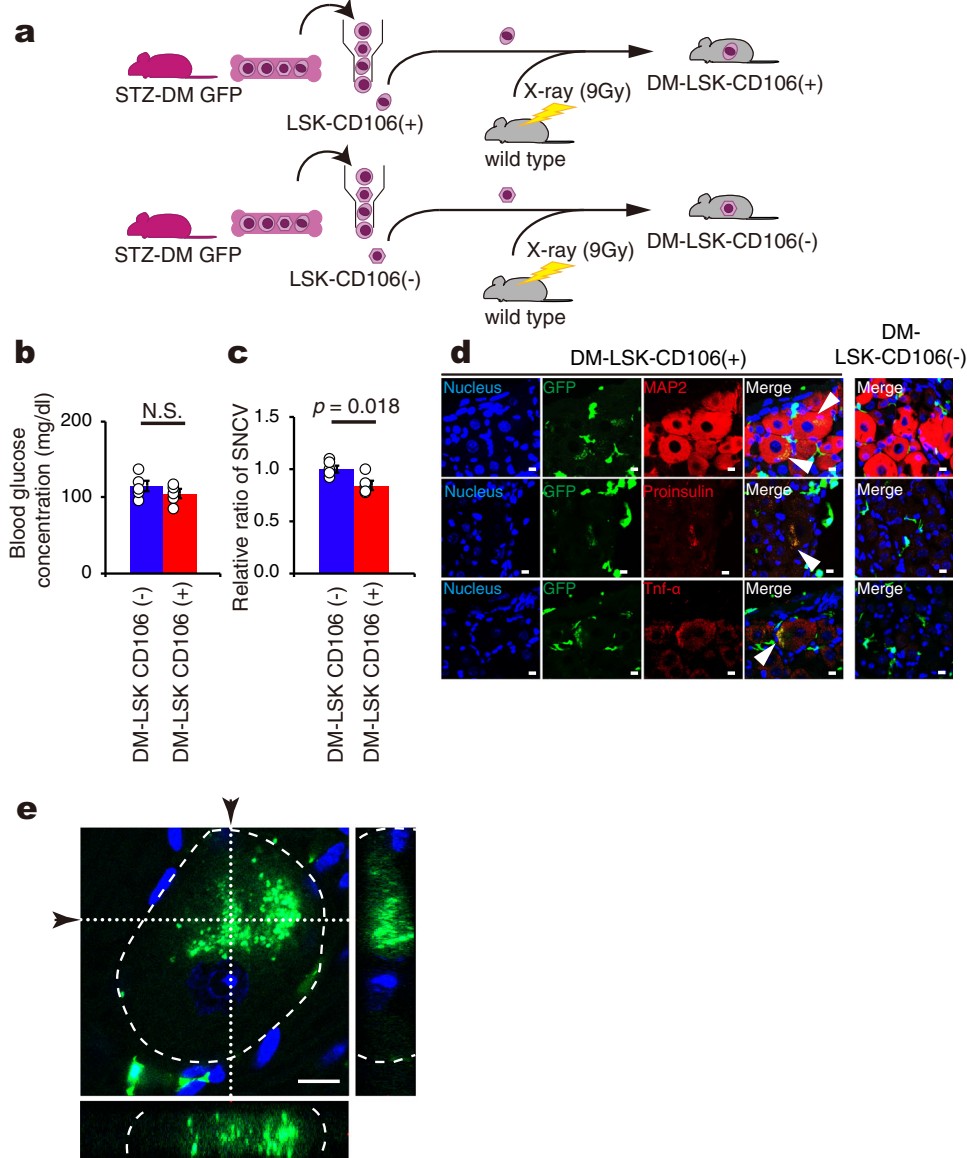

**Fig. 5 Cell fusion and development of neuropathy in mice transplanted with LSK-CD106-positive cells from diabetic mice. a** A schematic experimental design for the generation of mouse models in this experiment. First, type 1 DM was induced in GFP-Tg mice by intravenous STZ injection (STZ-DM GFP). Three months later, 9 Gy lethally-irradiated normoglycemic wild type mice were transplanted with LSK-CD106-positive or -negative cells obtained from STZ-DM GFP (DM-LSK-CD106 ( + ) BMT, DM-LSK-CD106 ( − ) BMT, respectively). **b** Blood glucose concentration of DM-LSK-CD106( + ) BMT (n = 5) and DM-LSK-CD106 ( − ) BMT (n = 5). N.S means not significant. **c** Relative ratio of sensory nerve conduction velocity (SNCV) of the sciatic nerve in DM-LSK-CD106 ( + ) BMT (n = 5) and DM-LSK-CD106 ( − ) BMT (n = 5). **d** Immunofluorescent staining of dorsal root ganglia (DRG) in DM-LSK-CD106 ( + ) BMT and DM-LSK-CD106 ( − ) BMT and for the nucleus (blue), GFP (green), MAP2 (red), proinsulin (red), and TNF-α (red), and merged images. Arrowheads showed fusion cells. **e** Z-stack imaging of DRG from DM-LSK-CD106( + ) BMT. Narrow images indicate the cross-section of x-axis or y-axis (whited dotted line) in DRG neuron surrounded by white dashed line. Arrows indicate the position of cross-section. Green showed GFP. Blue showed nucleus. Scale bar = 10 μm. Data are indicated as means ± SE.

Next, a new transplantation model was created to investigate whether CD106 cells emerging in diabetic bone marrow would be a new therapeutic target (Fig. 6). We isolated CD106-positive or-negative mononuclear cells by Ficoll and FACS methods from both nondiabetic and diabetic mice and transplanted those cells to normoglycemic mice (Fig. 6a). As expected, the SNCV is significantly delayed only the mice transplanted DM-CD106-positive mononuclear cells compared with the other recipient mice (Fig. 6b) though blood glucose concentration of all recipient mice is within normal range (Fig. 6c). Because the mice transplanted the CD106-negative mononuclear cells obtained

from both nondiabetic and diabetic mice were alive, we thought that the elimination of CD106-positive fraction from the diabetic hematopoietic stem cells is the novel strategy for the treatment of diabetic neuropathy.

Moreover, we examined CD106 antibody treatment for diabetic mononuclear cells to treatment diabetic neuropathy (Fig. 7a). The SNCV of the transplanted mice after the treatment with diabetic mononuclear cells recovered to the same level as the transplanted mice after non-diabetic mononuclear cell treatment and the transplanted mice after non-diabetic mononuclear cell treatment. On the other hand, SNCV of the transplanted mice

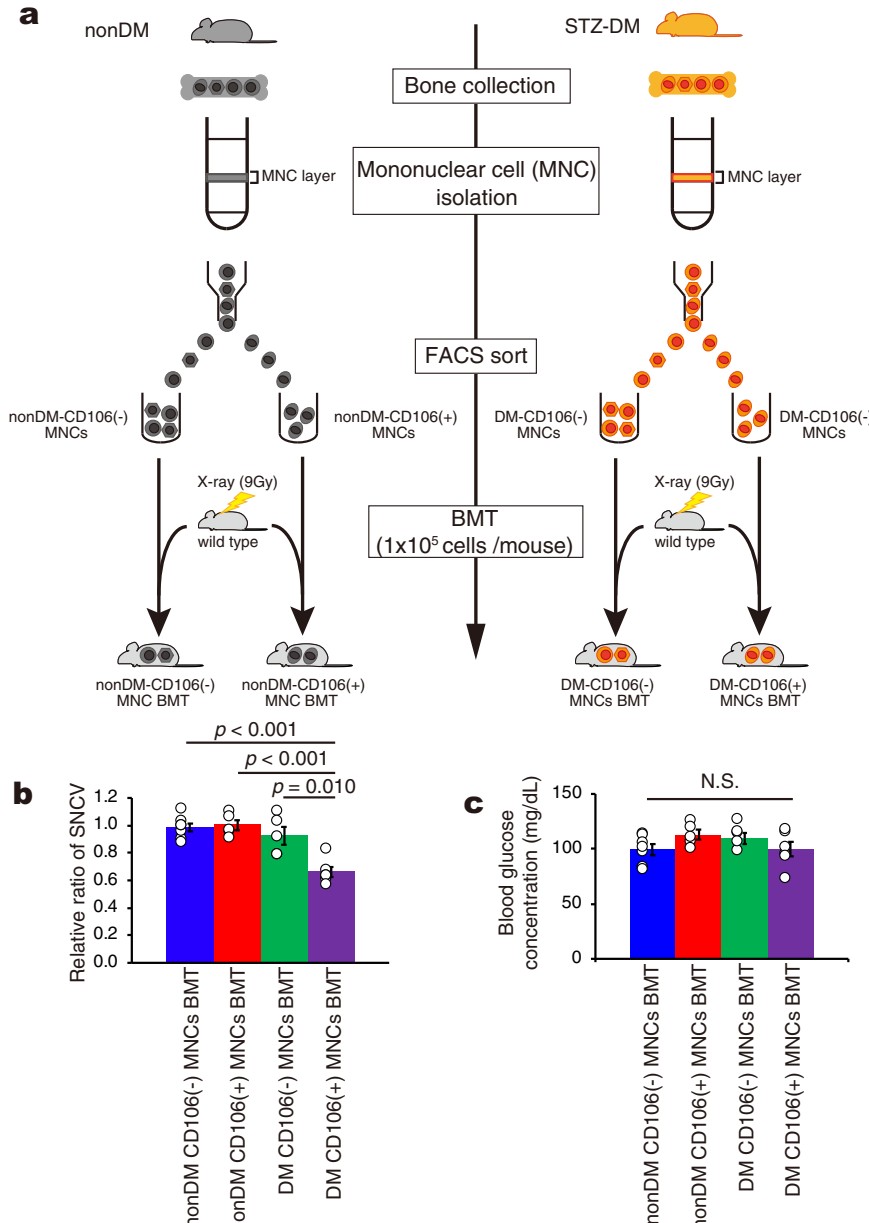

**Fig. 6 Development of neuropathy in mice transplanted with CD106-positive mononuclear cells from diabetic mice.** We isolated mononuclear cells using ficoll-paque PLUS from nondiabetic and diabetic total bone marrow. The isolated mononuclear cells were immunoreacted by PE-conjugated anti-CD106 antibody and sorted CD106-positive or -negative mononuclear cells by FACSAria fusion. Before anti-CD106 antibody staining, LIVE/DEAD Fixable Dead Cell Blue stain kit was used to deplete dead cells. Sorted cells (1 ×10⁵ cells / mouse) were transplanted into 9 Gy lethally-irradiated normoglycemic mice. Four weeks after BMT, we measured blood glucose and sensory nerve conduction velocity. **a** A schematic experimental method for CD106-positive or -negative mononuclear cells transplantation model. **b** Relative ratio of SNCV of sciatic nerve in nonDM CD106(−) MNCs BMT ($n = 7$) compared to that in nonDM CD106(+) MNCs BMT ($n = 5$), DM CD106(−) MNCs BMT ($n = 5$) and DM CD106(+) MNCs BMT ($n = 6$). **c** Blood glucose concentration of nonDM CD106(−) MNCs BMT ($n = 7$), nonDM CD106(+) MNCs BMT ($n = 5$), DM CD106(−) MNCs BMT ($n = 5$) and DM CD106(+) MNCs BMT ($n = 6$). N.S means not significant. Data are indicated as means ± SE.

after untreated diabetic mononuclear cells was significantly delayed (Fig. 7b). In addition, glucose concentration of recipient mice was all within normal range (Fig. 7c).

**Transient hyperglycemia induces neuropathy with cell fusion.** Our data showing that the abnormal BMDCs formed in response to diabetic conditions are capable of evoking neuropathy despite being under prolonged normoglycemia suggest that the insults caused by transient diabetic conditions could be memorized in

BMDCs, and such memory could be associated with TNF-α/ proinsulin expression in CD106-positive stem cells. Among multiple metabolic abnormalities in diabetic conditions, we assumed that hyperglycemia could be the factor that induced the memory effects in BMDCs. In order to address this issue, we designed an experiment in which repeated transient hyperglycemic surges were induced in wild type mice with BMT from GFP mice by repeated intraperitoneal injection of 25% glucose solution for 14 days while saline-injected mice were used as controls (Fig. 8a)[21]. We confirmed that glucose injection four times a day

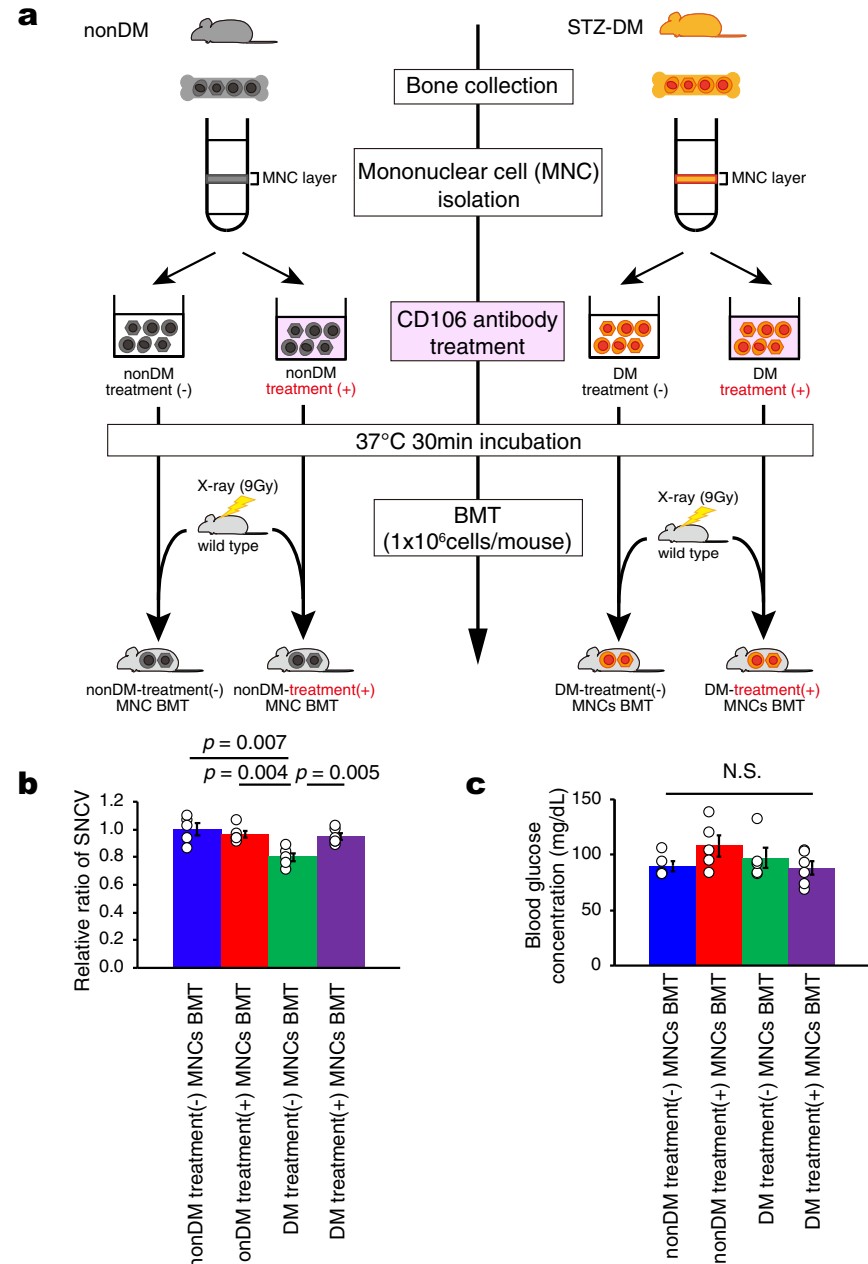

**Fig. 7 Treatment of anti-CD106 antibody to diabetic mononuclear cells was improved diabetic neuropathy.** We isolated mononuclear cells using ficoll-paque PLUS from nondiabetic and diabetic total bone marrow. The isolated mononuclear cells were incubated anti-CD106 antibody (final concentration is 10 μg/ml) for 30 min at 37 degree. Antibody-treated and nontreated cells (1 ×10⁶ cells / mouse) were transplanted into 9 Gy lethally-irradiated normoglycemic mice. Four weeks after BMT, we measured blood glucose and sensory nerve conduction velocity. **a** A schematic experimental method for CD106 antibody treatment (+) or (−) mononuclear cells transplantation model. **b** Relative ratio of SNCV of sciatic nerve in nonDM treatment (−) MNCs BMT ($n = 5$), nonDM treatment (+) MNCs BMT ($n = 5$), DM treatment (−) MNCs BMT ($n = 5$) and DM treatment (+) MNCs BMT ($n = 6$). **c** Blood glucose concentration of nonDM treatment (−) MNCs BMT ($n = 5$), nonDM treatment (+) MNCs BMT ($n = 5$), DM treatment (−) MNCs BMT ($n = 5$) and DM treatment (+) MNCs BMT ($n = 6$). N.S means not significant. Data are indicated as means ± SE.

significantly increased blood glucose concentration to as much as 400 mg/dL whereas saline injections remained it below 200 mg/dL in mice (Fig. 8b). Interestingly, repeated episodic glucose injections decreased SNCV by 14% compared with saline injection, indicating that relatively short-term pulsative hyperglycemia for 14 days was sufficient time for the induction of neuropathy even under non-diabetic conditions (Fig. 8c). We next examined whether the cell fusion process was involved in neuronal dysfunction in these mouse models. We found the presence of bone marrow-derived GFP-positive cells (green) at DRG sites in both (Fig. 8d); however,

proinsulin- (red) or TNF-α-expressing (red) cells were induced at DRG neurons only in mice receiving episodic glucose injections (Fig. 8d middle and bottom panels). Although MAP2 expression (red) was seen in all neurons of both models, only glucose injection induced positive signals for both MAP2 and GFP (green) in some DRG neurons (Fig. 8d top panels). Furthermore, both proinsulin- (red) and TNF-α (red) were induced in GFP-positive (green) cells of mice receiving episodic glucose injections (Fig. 8d middle and bottom panels). Furthermore GFP-positive signal was expressed inside DRG neuron (Fig. 8e). These results indicate that hyperglycemia due to

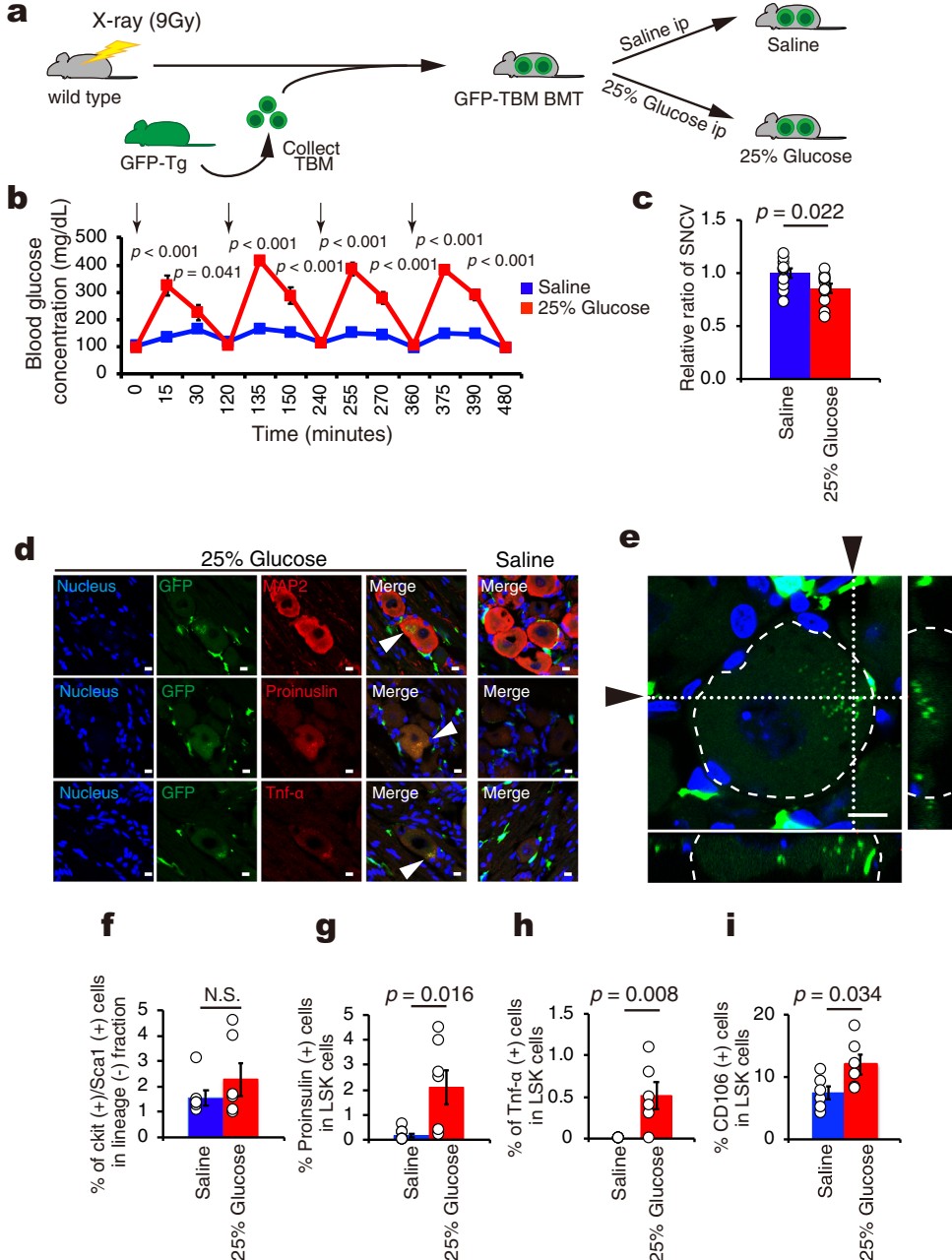

**Fig. 8 Cell fusion and development of diabetic neuropathy in wild mice injected intraperitoneally with 25% glucose. a** A schematic experimental design for the generation of 25% glucose- (25% Glucose) or saline- (Saline) injected wild type mice transplanted total bone marrow cells (TBM) from GFP-Tg (GFP-TBM BMT). ip, intraperitoneal injection. **b** The time course of blood glucose concentrations in GFP-TBM BMT mice treated with repeated intraperitoneal injection four times every two hour for 14 days. Arrows indicate timings of intraperitoneal injections (Saline $n = 11$, 25% Glucose $n = 12$). Red line indicates the effect of 25% glucose injection whereas blue line shows that of saline. **c** Relative ratio of SNCV of sciatic nerve in Saline ($n = 11$) compared to that in 25% Glucose ($n = 12$). **d** Immunofluorescent staining of dorsal root ganglion (DRG) for nucleus (blue), GFP (green), MAP2 (red), proinsulin (red), and TNF-α (red), and merged figures. Left panel shows the effects of 25% Glucose and right panel shows that of Saline. Arrowheads showed fusion cell. Scale bars=10 μm. **e** Z-stack imaging of DRG from 25% Glucose. Narrow images indicate the cross-section of x-axis or y-axis (whited dotted line) in DRG neuron surrounded by white dashed line. Arrows indicate the position of cross-section. Green showed GFP. Blue showed nucleus. Scale bar = 10 μm. **f** The effect of injection of either Saline or 25% Glucose on the percentage of LSK cells in mononuclear cell fraction from total bone marrow. Percentage of LSK cells in the mononuclear cell fraction from total bone marrow in Saline and 25% Glucose ($n = 6$ per group). **g** Percentage of proinsulin-positive cells in LSK cells from Saline and 25% Glucose ($n = 7$ per group). **h** Percentage of TNF-α-positive cells in LSK cell fraction from mononuclear cell in Saline and 25% Glucose ($n = 6$ per group). **i** Percentage of CD106-positive cells in LSK cell fraction from mononuclear cell ($n = 6$ per group). Data are indicated as means ± SE.

episodic glucose injections is sufficient to induce cell fusion between BMDCs and neurons.

To clarify whether hyperglycemia itself induces the appearance of abnormal HSCs, we examined whether repeated episodic

glucose injections could alter the phenotypes of bone marrow cells in nonDM mice. Even though there was no change in the percentage of LSK cells (Fig. 8f), we found that repeated episodic glucose injections significantly induced proinsulin expression in

LSK cells, but saline injections had very minor effects (Fig. 8g). Interestingly, episodic saline injections were also found to slightly induce proinsulin expression in LSK cells although the numbers of such cells were extremely low. The reason might be accounted for by mechanical stress due to repeated intraperitoneal saline injection. With respect to TNF-α expression, positive staining was observed in 0.5% of LSK cells, but no expression was detected in the saline model (Fig. 8h). In turn, CD106-positive staining was present in 7% of LSK cells in mice given saline injection but increased to 12% of LSK in mice given glucose injections. The difference in the number of CD106-positive cells reached statistical significance ($P = 0.034$) (Fig. 8i). Next, we divided LSK cells into two fractions (SP and nonSP), and examined the expression of TNF-α and CD106. Glucose injections failed to alter the population of SP and nonSP compared with saline injection (Supplementary fig. 4a). CD106-positive cells were present only in the nonSP fraction, but not in the SP fraction, in both models. The percentage of CD106-positive cells in the nonSP fraction was increased by glucose injections compared with saline injections (Supplementary Fig. 4b, c). With respect to TNF-α expression, we found that no cells were positive in the LSK-SP fraction of either model while 0.2% of LSK cells in the nonSP fraction of mice given episodic glucose injections expressed TNF-α (Supplementary Fig. 4d, e). These results indicate that hyperglycemia could be a mechanism for TNF-α induction in the CD106-positive fraction of ST-HSCs. Taken together, sustained hyperglycemia over several days appears sufficient to induce the expression of proinsulin and TNF-α expression in ST-HSCs, transforming those cells into the major villain that causes diabetic neuropathy. These specific cells would therefore be a good therapeutic target for the treatment of diabetic neuropathy.

## Discussion

While DN is believed to be caused by diabetes-associated metabolic abnormalities in neurons, current therapies aiming at glycemic control appear inadequate as curative tools for DN. Clearly, new hypotheses and approaches are warranted. We previously reported that a potential mechanism for DN could be "cell fusion" occurring between neurons and abnormal BMDCs, originating among HSCs[13,18]. In this study we delved further into the mechanisms involved and demonstrated that CD106-positive LSK cells which emerged in bone marrow under hyperglycemia were instrumental in inducing cell fusion in DRG neurons, leading to neural dysfunction even in nondiabetic mice. These results clearly indicate that DN is caused by abnormal BMDCs, not simply by hyperglycemia-associated metabolic alterations in nerve tissues.

DRG neurons do not physiologically express TNF-α/proinsulin under nonDM conditions[22,42]. However, we demonstrated in this study that diabetic conditions force these cells to start producing TNF-α/proinsulin, suggesting that TNF-α/proinsulin expressions do not mean the trans-differentiation process but indicates a consequence of cell fusion with abnormal BMDCs. In other words, diabetic conditions initially perturb the functions of bone marrow cells, which then move into the systemic circulation to reach the DRG and liver[13,14]. At those sites, abnormal BMDCs fuse with resident cells to adversely affect the physiological function of the resident cells. As such, the true culprit for the development of DN is functionally-impaired BMDCs as these cells express TNF-α/proinsulin in response to diabetic conditions and cause "cell fusion" with DRG neurons. In this study, the pathological role of diabetic bone marrow cells was also examined in an experiment using the transplantation of bone marrow cells from DM mice into nonDM mice under normoglycemic condition. This showed that diabetic bone marrow cells expressing

TNF-α/proinsulin cause cell fusion in nonDM mice even under normoglycemic conditions. Among several metabolic abnormalities in diabetes, hyperglycemia per se impair BMDC functions most potently as simple glucose injection into the peritoneal reproduced the same phenotype in BMDCs as diabetic mice. Recently, several researchers have shown interesting findings that diabetes could impair the mobilization of hematopoietic stem cells and myelopoiesis as a mechanism for diabetic complications[43,44]. The future study is warranted to investigate if HSC we found could be associated with abnormal myelopoiesis.

More importantly once established, the functional impairment of BMDCs does not return to normal even if blood glucose control is significantly improved. This explains why these cells can cause neuronal dysfunction in nonDM mice. It is likely that diabetes can cause a lasting disruption of BMDC function, which may not be immediately reversible upon restoration of normoglycemia. These data could account for the fact that current therapeutic strategies aiming at glycemic control are often not effective in treating DN in diabetic patients.

To further identify the true culprit for DN, we employed microarray analysis and found increases in gene expression involving multiple factors in diabetes-specific stem cells. Among the genes affected, we focused on the CD106- (Vcam1) positive cell population as a possible candidate because CD106 is generally expressed in the pathological process under diabetic conditions[45] and in cancer angiogenesis[46]. To investigate the role of CD106, LSK-CD106-positive cells and LSK-CD106-negative cells were transplanted separately into nonDM mice. As expected, some TNF-α-positive cells were found only in mice receiving LSK-CD106-positive cells but not those receiving LSK-CD106-negative cells. The findings suggest that those CD106-positive HSCs expressing TNF-α exhibited phenotypic change and therefore are a candidate for diabetes-specific stem cells. Interestingly, the pathological role of CD106-positive cells is also supported by other studies showing that these cells are associated with chronic inflammation and metastatic cancer[47–49]. Furthermore, both integrin-alpha4 and -beta1 are expressed in DRG neurons, the increased Vcam1 expression in diabetic LSK cells may in part play an important role in neuronal cell fusion as indicated in Supplementary fig 3. In addition, abnormal endothelial formation may also have roles in diabetes[36,50].

It is reported that CD106 does not only express on inflammation-induced endothelial cells, but also participate in hemopoiesis, B cell development and T cell activation[51]. These reports indicate that CD106 have also important role in immunological reaction. However, the model mice transplanted with nonDM CD106(-) MNCs and STZDM CD106(-) MNCs survived without weight loss or serious infections. These data indicate that even in the diabetic state, LT-HSCs without CD106-positive cells retain their stem cell integrity and have normal differentiation potential. Thus, hyperglycemia-induced abnormal ST-HSCs are common progenitor cells with abnormal differentiation potential for fusion with peripheral neurons, which are specific for diabetes and never appear in non-diabetes.

In turn, we also found that CD106-negative cells in diabetic mice have no pathological roles in the development of DN although those cells are also present among diabetic BMDCs. This finding was also confirmed by an experiment in which hyperglycemia was induced by episodic intraperitoneal glucose injections over several days. Based on our results, eliminating abnormal HSCs such as CD106-positive cells would be a novel way to treat DN[52].

## Methods

**Animal model.** The Animal Care Committees of Shiga University of Medical Science approved all experimental protocols in this study. C57BL/6 J mice (wild

type, Japan SLC, Inc., Shizuoka, Japan), C57BL/6-Tg (UBC-GFP) 30Scha/J mice (GFP-Tg mice, The Jackson Laboratory, Bar Harbor, ME, USA), B6.Cg-Gt(ROSA) 26Sor[tm9(CAG-tdTomato)Hze]/J (Lsl-tdTomato mice, The Jackson Laboratory) and Ayu1 promoter-driven Cre recombinase-expressing mice[26] (Ayu1-Cre) were used. We examined two types of diabetic mice. One was a type 1 diabetic mouse model created by injecting streptozotocin (STZ: 150 mg/kg Body Weight) (Nacalai Tesque, Kyoto, Japan) into wild-type C57BL/6 J mice. The other was a type 2 diabetes mouse model produced by feeding wild-type C57BL/6 J mice a high fat diet (HFD) (Oriental Yeast Co., Ltd., Tokyo, Japan). The nondiabetic control group for Type 1 mice received citrate buffer (pH 4.5) injection; control mice for the type 2 model were fed normal chow. The blood glucose concentrations of all mice were measured every week after injection of either STZ or buffer and feeding with a HFD or normal chow. Mice were maintained under standard conditions with a 12:12-h light-dark cycle and provided with standard laboratory chow or HFD and water ad libitum.

First, we performed experiments using two diabetic mouse models. Type1 DM was induced by streptozotocin (STZ) treatment (STZ-DM) (Supplementary Fig. 1a). Type 2 DM was induced by high-fat-diet treatment (HFD-DM) (Supplementary Fig. 1e). To monitor cell fusion, we created a new mouse model, in which TBM from Ayu-1 Cre transgenic mice, which had a gene modification to develop Cre recombinase from the embryonic stage[26], was transplanted into LoxP-stop-LoxP-tdTomato (lsl-tdTomato) mice, in which the tdtomato gene had been inserted downstream of the STOP sequence inserted in the LoxP site (Ayu-1 TBM BMT to lsl-tomato) (Fig. 1a). Type1 DM was induced by STZ-injection in this model.

The second series of experiments was performed on a nondiabetic mouse model that received transplantation of total bone marrow (approximately $1 \times 10^6$ cells /mouse) obtained from diabetic mice. The bone marrow-donor mouse (GFP-Tg) received STZ injection before bone marrow collection. Donor cells were transplanted into nondiabetic recipient mice that had received 9 Gy radiation before BMT (Supplementary Fig. 2a) as described previously[19].

In the third series of experiments, we used nondiabetic recipient mice that had received LSK fraction hematopoietic stem cells (HSCs) collected from diabetic mice (Fig. 2a). Mononuclear cells were isolated from total bone marrow cells using Ficoll-Paque Plus (GE Healthcare Bio-Sciences AB, Uppsala, Sweden), and stained with LIVE/DEAD Fixable Blue Dead cell stain kit (Thermo Fisher Scientific Inc., Waltham, MA, USA) to selectively deplete dead cells. The isolated mononuclear cells were washed with phosphate buffered saline (-) (PBS (-)), suspended in Flow Cytometry Staining Buffer (eBiosciences, Inc., San Diego, CA, USA), and stained with a Biotin Mouse Lineage Panel that includes the following antibodies: anti-CD3e, anti-CD11b, anti-CD45R, anti-Gr-1, and anti-TER119 (BD Biosciences, San Jose, CA, USA), followed by fluorescent antibodies: Phycoerythrin (PE)-Cy7-conjugated streptavidin, allophycocyanin (APC)-conjugated anti-c-kit, and APC-Cy7-conjugated anti-Ly6A/E (Sca-1) (all antibodies were purchased from BD Biosciences). Finally, LSK cells were sorted on a FACS Aria Fusion (BD Biosciences). The isotype-matched control antibodies were used to define the gate threshold of specific antibodies. The lineage-negative gate was obtained by the addition of PE-Cy7 streptavidin-conjugated antibody. Approximately $1 \times 10^3$ LSK cells were then transplanted into lethally-irradiated wild-type mice (called LSK-T from nonDM or LSK-T from STZ-DM, respectively). Weekly measurement of blood glucose concentration at 9:00 AM was started at 4 weeks. After LSK cells were sorted according to the expression of CD106 in diabetic mice, either LSK-CD106-positive cells or LSK-CD106-negative cells were transplanted into nondiabetic mice to create two types of mouse models; DM-LSK-CD106 ( + ) BMT or DM-LSK-CD106 (-) BMT, respectively (Fig. 5a).

Finally, we performed experiments using nondiabetic mice with glucose-injection induced hyperglycemia. The nondiabetic mice received BMT (approximately $1 \times 10^6$ total bone marrow cells/100ul /mouse) from nondiabetic GFP-Tg mice and received intraperitoneal injections of 25% glucose solution (3 g/kg Body Weight) or saline four times a day for 14 days (Fig. 6a). Blood glucose concentration was measured at 4 time-points (0, 15, 30 and 120 minutes) after each injection.

To examine whether the diabetic CD106-positive cells are novel therapeutic target, we made two transplantation models, first: the normoglycemic mice that received the transplantation of CD106-positive or -negative mononuclear cells obtained from nondiabetic and diabetic bone marrow (Fig. 7), second: the normoglycemic mice that received the transplantation of mononuclear cells obtained from nondiabetic and diabetic bone marrow with or without anti-CD106 antibody treatment (Fig. 8). Anti-CD106 antibody (biolegend Inc., Clone 429 (MVCAM.A), final concentration:10 μg/ml) was utilized in order to neutralize the adhesive action of CD106 mononuclear cells at DRG neurons.

**Determination of blood chimerism after BMT.** To examine the re-establishment of the hematopoietic system, peripheral blood was taken from the saphenous vein, and analyzed by FACS to determine the proportion of GFP-positive cells in peripheral blood at 12 weeks after transplantation of total bone marrow, LSK cells, or LSK-CD106-positive cells using a FACS Canto II and FACS DIVA software (BD Biosciences). We selected only mice with more than 90% of peripheral white blood cells of donor origin for further study.

**Sensory nerve conduction velocity.** In order to expose the sciatic nerve, the skin was cut at the left dorsal surface of the thigh to measure sensory nerve action potential (SNAP) using a Medelec Sapphire electromyograph (Medelec, Woking, UK) under anesthesia at 30 to 32 °C (13, 18), and sensory nerve conduction velocity (SNCV) was calculated.

**Immunofluorescent staining of the dorsal root ganglion.** After exsanguination, mice were perfused with fixative to collect a dorsal root ganglion (DRG) as previously described[13,18]. Eight μm-thick frozen sections were immunostained with the following antibodies: rabbit anti-MAP2, rabbit anti-insulin (Cell signaling technologies, Danvers, MA USA) or rabbit anti-TNF-α (Abcam plc. Cambridge, UK) as primary antibodies, followed by Alexa555 anti-rabbit IgG antibody as the secondary antibody (Thermo Fisher Scientific). Finally, sections were mounted with Vectashield mounting medium with DAPI (Vector Laboratories, Burlingame, CA, USA), and then analyzed by confocal laser-scanning microscopy (CLS, EZ-C1 software, Nikon, Tokyo, Japan). TdTomato-fluorescence expression was detected by histological analysis. NeuroTrace Fluorescent Nissl stain (Thermo Fisher Scientific) was used for the detection of DRG neurons.

**Expression of TNF-α, proinsulin, and CD106 by LSK cells.** To evaluate TNF-α and CD106 expression, we isolated mononuclear cells from total bone marrow using Ficoll-Paque Plus with PE-Cy7-conjugated streptavidin antibody, followed by immunoreaction with APC-conjugated anti-c-kit, APC-Cy7-conjugated anti-Ly6A/E (Sca-1), fluorescein isothiocyanate (FITC) -conjugated anti-CD106 or FITC-conjugated anti-CD45 and phycoerythrin (PE)-conjugated anti-TNF-α (eBiosciences) after Biotin Mouse Lineage Panel staining. To evaluate proinsulin expression, mononuclear cells were fixed with BD Cytofix/CytoPerm (BD Biosciences) after reacting with PE-Cy7-conjugated streptavidin antibody, APC-conjugated anti-c-kit, APC-Cy7-conjugated anti-Ly6A/E (Sca-1), and Biotin Mouse Lineage Panel. Then a rabbit anti-insulin monoclonal antibody (Cell signaling technology) and PE-conjugated anti-rabbit IgG antibody (Cell signaling technology) were applied to mononuclear cells. To deplete dead cells, LIVE/DEAD Fixable Dead Cell Blue stain kit was used before staining. Four hours after staining, cells were analyzed using a FACS Canto II with FACS DIVA software (BD Biosciences).

**Genomic DNA analysis.** In order to confirm deletion of the stop sequence in DRG neurons as a consequence of cell fusion, genomic DNA was extracted from DRG neurons in either diabetic or nondiabetic condition of Ayu-1 Cre TBM lsl-tomato mice by NucleoSpin DNA FFPE XS kit (MACHEREY-NAGEL GmbH & Co. KG, Düren, Germany), and PCR was performed by TaKaRa Ex Taq (TaKaRa BIO INC., Kusatsu, Shiga, japan). Following primers were used: Forward: 5′-GCTGATC CGGAACCCTTAA-3′ and Reverse: 5′-CCCATGGTCTTCTTCTGCAT-3′.

**Microarray analysis.** RNA was isolated from LSK cells using a RNeasy Plus Micro kit (Qiagen, Tokyo, Japan). The RNA samples were analyzed by TaKaRa bio (Shiga, Japan) using a Nanodrop and Agilent 2100 BioAnalyser. The RNA samples were amplified by the Ovation Pico WTA system, and used to perform microarray analysis.

**Mesenchymal stem cell analysis.** We performed FACS analysis to characterize the bone marrow mesenchymal stem cells obtained from diabetic and nondiabetic mice. For the preparation of nucleated cells, total bone marrow cells were collected and hemolyzed in ammonium chloride-based red blood cell lysis buffer for 60 s, then washed twice with PBS(-). Nucleated cells were stained with PE-Cy7-conjugated anti-CD45 antibody (BioLegend Inc., San Diego, CA, USA), APC-Cy7-conjugated anti-Ly6A/E (Sca1), APC-conjugated anti-CD105 (BioLegend Inc.), or FITC-conjugated anti-CD106. Before specific antibody staining, LIVE/DEAD Fixable Dead Cell Blue stain kit was used to deplete dead cells. Finally, a FACS Canto II with FACS DIVA software (BD Biosciences) was used for data collection and analysis.

**Analysis of the side population in LSK cells.** The side population (SP) of LSK cells was analyzed by FACS according to the method reported by Goodell[53]. Total bone marrow cells were stained with Hoechst33342 dye (Sigma-Aldrich Japan K.K., Tokyo, Japan) for exactly 90 min at 37 °C. Hoechest33342-positive cells were incubated with verapamil hydrochloride (Tocris bioscience, Bristol, UK) for gate setting, and then mononuclear cells were separated by Ficoll-Paque Plus. Next, these cells were incubated with PE-Cy7-conjugated streptavidin, APC-conjugated anti-c-kit, APC-Cy7-conjugated anti-Ly6A/E (Sca-1) and PE-conjugated anti-TNF-α or FITC-conjugated anti-CD106 after Biotin Mouse Lineage Panel staining. Finally, for dead cell depletion, propidium iodide (Sigma-Aldrich) was added to the samples to remove dead cells before analysis using a FACS Aria Fusion with FACS DIVA software (BD Biosciences).

**Statistical analysis.** In this study, we performed unpaired student's t-test to evaluate between 2 groups or ANOVA among 4 groups. All data are showed as means ± SE and $P < 0.05$ indicated significantly differences.

**Statistics and Reproducibility**. For statistical analysis of the results, *t*-test was performed between the two groups, and one-way ANOVA Tukey's post hoc analysis was performed between the four groups. The *t*-test was performed using Microsoft Excel, and one-way ANOVA was performed using SPSS software version 22 (IBM, Armonk, New York, USA).

**Reporting summary**. Further information on research design is available in the Nature Research Reporting Summary linked to this article.

## Data availability

All raw data that were graphed are provided as Supplementary data. The microarray results of this study have been deposited in the Gene Expression Omnibus (GEO) with Accession number is GSE117088. All other data are available from the corresponding author on reasonable request.

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

## Acknowledgements
This work was supported by a Grant-in-Aid (No. 16K19051 to M.K., No. 16659240, 18390100 and 23590378 to H.K., No. 17659264 to T.T., and No. 16K15756 to Y.N.) for Scientific Research from the Ministry of Education, Culture, Sports, Science and Technology, Japan, and by the President's Discretionary Fund from Shiga University of Medical Science (No. 1515503 W, 1515503ZE and 1515503ZB for M.K., and No. 1515503ZZE for H.K.). We thank Dr. Chan L. for his great effort in discussing our research and writing our paper, and M. Kanaya and M. Kunimatsu for genotyping and maintenance of the mice, and Y. Mori and T. Yamamoto for technical supports in FACS analyses and bone marrow transplantations.

## Author contributions
M.K. and H.K. performed most of the experiments, analyzed the data and prepared it for presentation, and contributed to experimental design and manuscript writing. T.T. contributed to in vivo experiments, design of the experiments, review of data and manuscript editing. Y.N. assisted with in vivo experiments and immunohistochemistry experiments. A.Y. assisted with immunohistochemistry experiments. H.M contributed to in vivo experiments and manuscript editing. H.K., T.N, N.O., I.M., J.O., Y.S., K.F., and Y. E. contributed to design of the experiments, review of the data and manuscript editing.

## Competing interests
All authors declare no competing interests.
