## [Peer Review File · Communications Biology]

Reviewers' comments:

Reviewer #1 (Remarks to the Author):

This is a very interesting study attempting to demonstrate that a subpopulation of hematopoietic cells participates in the pathogenesis of diabetic neuropathy. The authors use different animal models and confirmed that a specific population expressing CD106 fuse with ganglionic cells. When transplanted in non-diabetic mice or when non-diabetic mice are exposed to hypoglycemia, these cells reproduce the characteristics of neuropathy. They conclude these bone marrow cells are responsible for diabetic neuropathy and should be a target for future treatments.

As most of the interesting papers, this one raises many questions.

1. Diabetic neuropathy is a multifactorial disease, do the authors really think this can be only reconducted to the reported cellular phenomenon?
2. If hyperglycemia is the cause of this mechanism, does control of high glucose reduce neuropathy and cell recruitment to the ganglia?
3. What is the mechanism this subpopulation emerges in diabetic bone marrow? Do these LSK cells upregulate CD106 under high glucose or oxidative stress? Are they more propense to apoptosis?
4. What is the mechanism of recruitment to ganglia? Our group has shown that neuropeptide substance P is reduced in diabetic mice thus causing a defect in regenerative cells recruitment upon peripheral injury (Amedesi, Circulation 2012). The authors should try to demonstrate what is the chemokine signalling attracting these cells to peripheral nerves in diabetes. Why are only these cells recruited?
5. From a translational standpoint, it would be important to determine if the frequency of CD106 hematopoietic cells in the circulation is associated with the severity of neuropathy in diabetic patients. This is an important point to confirm that the observed phenomenon has clinical relevance.
6. The authors should induce selective depletion of the CD106 fraction to demonstrate these cells are the only determinants of neuropathy. They mention these cells are a possible target for therapy. However, if they cannot be depleted or instructed to behave as normal cells, the statement remains unconfirmed.

Reviewer #2 (Remarks to the Author):

In this interesting manuscript by Dr. Kojima and collaborators, a novel cellular mechanism responsible for the development of diabetic neuropathy in STZ-treated mice was identified. Cell fusion between aberrant CD106-positive BMCs and dorsal root ganglion cells appears to be the process by which diabetic neuropathy develops.

The occurrence of bona fide fusion between CD106-positive BMCs and dorsal root ganglion cells was documented elegantly in a genetically engineered mouse. The transplantation experiments are well done and convincing. The expression of proinsulin, however, is indicative of a process of transdifferentiation that remains unexplained and should be taken into account. The ability of these cells to form beta-like cells should be explored.

Reviewer #3 (Remarks to the Author):

Katagi and colleagues describes a subpopulation of HSC that express the antigen CD106, which can be transplanted and can engraft a recipient bone marrow. Yet, the progeny originating from these cells (?) appear to fuse with DRG cells and trigger a neuronal dysfunction that the authors associate with a form of diabetic neuropathy.

-The main concern regards the quality of the flow cytometry data. In figure 2e, 3e, 6f and 6g the authors showed the expression of proinsulin and TNF alpha on LKS cells. It is known, and for sure to the authors as well, that LKS cells are a small percentage of the BM cells. Therefore, to show convincingly the expression of different markers (expressed a very low level) on a small population a large number of event is commonly required. The authors reported histogram plots that are far from convincing since they show a negligible percentage of positive cells (ranging from a 0.5 to 3%) with too few events. Those charts are unacceptable in their present form and are far from being convincing. The authors should increase substantially the number of events acquired to get more reliable data.

-Microarray data should be given with an accession number and also with the whole data set.

-In figure 3d non diabetic mice, have about 1% of LKS cells positive for CD106, while diabetic mice are about 2 fold increase. In figure 6h non diabetic mice have about 5% of LKS cells positive for CD106 while the injection with saline increased the expression of CD106 up to 10%. How the authors explain the difference between these two data? Is this an effect of bone marrow transplantation?

-LKS are a mixed population of hematopoietic stem and progenitors cells. Usually other markers are required to identify LT-HSC, such as CD150, CD48 FLK1 or CD34. The authors acknowledge that the expression of CD106 encompass a population of ST-HSC, which usually have a lower regenerative potential. Did this cells engraft and reconstitute normally the bone marrow of the hosts? Did the authors see any defect or bias in the reconstitution of the bone marrow?

-It would be very insightful to assess whether the transplantation of CD106+ and CD106- LKS from non diabetic mice have the same effect of the diabetic counterpart. With this experiments the authors could clearly confirm that CD106 identify putative LKS cells capable of cell fusion and capable of triggering neuronal dysfunction.

-The first 2 paragraph of the first result should be moved to the introduction

-Reference to supplementary figure 2a and 2b and c are wrong in the text.

-The authors should provide z-Stack images, with xyz projection of DRG to validate cell fusion. With the current images is difficult to assess whether cells are actually fused or just overlapping.

-Did the authors have made any hypothesis about what kind of cell can fuse with DRG? Are they macrophages? Hematopoietic stem cells? Some kind of undefined progenitors?

*Communications Biology has recently integrated its submissions with Dryad. To submit your data to Dryad, please use the link supplied during your manuscript submission. If no Dryad link was provided, please contact the editor.

A Dryad account is required for data submission; you may need to create an account and/or log in to begin. If the link provided does not appear to be working after 24 hours, please contact help@datadryad.org.

Once your data package is deposited, a unique and stable identifier (DOI) will be sent to you and to the journal for inclusion in the published article. For questions or feedback on the Dryad data submission process, please email help@datadryad.org.

To the author: Please provide your responses to the reviewers' comments in the table below. Highlighted text simply points out when specific requests are made by reviewers that we want to make sure are not missed. Editorial comments may be used to draw attention to specific points that we feel are especially important or that we do not feel are necessary in a revision. They may also be used to note when a comment is related to another comment by the same or another reviewer. However, we ask that you respond to all reviewer comments as thoroughly as possible. The referees' comments have been reproduced in full and entered into the below tables by the editor, though the order in which they are presented may have changed. Please feel free to modify this table as needed, but please do not edit the referee comments column in any way.

Author's response summary:

Reviewer #1

	Comment	Editorial comment	Author Response	Modification made in manuscript (give quotes and page numbers)
1	Given that diabetic neuropathy is a multifactorial disease, do the authors think that the effects of ST-HSCs are limited to the reported cellular phenomena?		We think that diabetic neuropathy is not a multifactorial disease but a bone marrow stem cell disease as we demonstrated that diabetic neuropathy was rescued by a selective depletion of abnormal hematopoietic stem cells from mononuclear cell fraction. Please see the detailed explanation in our response for question No.6.	
2	If hyperglycemia is the cause of this mechanism, does control of high glucose reduce neuropathy and cell recruitment to the ganglia?	Can insulin injection test this idea?	We have already reported that the insulin treatment suppressed the progression of diabetic neuropathy and decreased bone marrow-derived cell recruitment to the ganglion (FASEB J Terashima T.et al)	We now put the following comment in the discussion (Page 17 in the revision). "More importantly once established, the functional impairment of BMDCs does not return to normal even if blood glucose control is significantly improved. This explains why these cells can cause neuronal dysfunction in nonDM mice."

3	How does this subpopulation emerge in diabetic bone marrow? Do these LSK cells upregulate CD106 under high glucose or oxidative stress? Are they more prone to apoptosis?		We think that the bone marrow subpopulation in the LSK stem cell fraction abnormally expressing proinsulin and Tnf-alpha could emerge in response to high glucose. It is because simple glucose injections also generated the subpopulation of LSK stem cells. Since high glucose induces oxidative stress as the reviewer mentions oxidative stress could be another mechanism, but we did not examine the effect of oxidative stress in our study.	We now put the following sentence in the discussion (page17). “Among several metabolic abnormalities in diabetes, hyperglycemia per se impairs BMDC functions most potently as simple glucose injection into the peritoneal reproduced the same phenotype in BMDCs as diabetic mice.”
4	How are this subpopulation recruited to ganglia? Our group has shown that neuropeptide substance P is reduced in diabetic mice, causing a defect in regenerative cell recruitment upon peripheral injury (Amedesi, Circulation 2012). The authors should try to demonstrate what is the chemokine signaling that attracts these cells to peripheral nerves in diabetes.	This is an interesting question. But answering this question thoroughly will not be required for publishing this study. If you have some preliminary data and/or speculation, it will be great to discuss it in the Discussion.	A novel mechanism we found in this study is that high glucose impairs the function of LSK cells, which directly cause diabetic neuropathy. In this process, abnormal LSK cells should be attracted by some chemokines, and it could be the neuropeptide substance P as you mention. But we did not examine it. Instead, we performed microarray assay and found that Vcam1, a strong adhesion molecule in the advance of atherosclerotic plaque formation or cancer cell metastasis, is increased in abnormal hematopoietic	

			stem cells. Given the fact that both integrin-alpha4 and -beta1 are expressed in DRG neurons, and these molecules can bind Vcam1, then the increased Vcam1 expression in diabetic LSK cells (x 8) may in part play an important role in neuronal cell fusion as indicated in Supplemental fig 3. In addition, abnormal endothelial formation may also have roles in diabetic retinopathy and whether it work or not should be explored.	
5	To test the clinical relevance of this finding, it would be important to determine if the frequency of CD106 hematopoietic cells in the circulation is associated with the severity of neuropathy in diabetic patients.	Even if authors don't find such association, it does not mean that CD106 hematopoietic cells do not play a role in human diabetic neuropathy because it may not be the circulatory levels of CD106 hematopoietic cells that matter.	In this study, only mouse studies were conducted and no human studies were conducted. We agree with your comments and are currently conducting a human study.	
6	The authors should induce selective depletion of the CD106 fraction to demonstrate these cells are the only determinants of neuropathy. They mention these cells are a possible target for therapy. However, if they cannot be depleted or instructed to behave as normal cells, the statement remains unconfirmed.	It is important to get this experiment done.	In accordance with the editorial comment and reviewers comment, we transplanted mononuclear cell containing CD106-positive cells found in diabetic bone marrow and its depleted fractions into non-diabetic mice, respectively, and examined the development of neuropathy. Next, we	We have added these results (page12) and new Figures 7 in the paper.

			transplanted diabetic bone marrow without or with CD106 neutralizing antibody into non-diabetic mice and examined the development of neuropathy. In both experiments, mice transplanted with bone marrow mononuclear cells from which the CD106 fraction had been removed did not develop diabetic neuropathy. We have added these results as new Figures 7 and 8 in the paper.	
--	--	--	---	--

Reviewer #2

	Comment	Editorial comment	Author Response	Modification made in manuscript (give quotes and/or page numbers)
1	The process of trans-differentiation of CD106-positive BMCs into beta-like cells cannot be excluded yet and the ability of these cells to form beta-like cells should be explored.		We have created a new transplantation model to show that as a result of cell fusion, neurons retain the proinsulin expressed in bone marrow-derived cells. In this model, we show that cell fusion results in the expression of the TOMATO fluorescent dye, red, in DRG neurons. This indicates that it is not transdifferentiated.	We now put this comment in the discussion (Page 16) as follows; “we demonstrated in this study that diabetic conditions force these cells to start producing TNF- α /proinsulin, suggesting that TNF- α /proinsulin expressions do not mean the trans-differentiation process but indicates a consequence of cell fusion with abnormal BMDCs”

Reviewer #3

	Comment	Editorial comment	Author Response	Modification made in manuscript (give quotes and/or page numbers)
1	Given that LKS cells are a small percentage of the BM cells, a large number of events are commonly required to show convincingly the expression of different markers on a small population in flow cytometry.	Please address this concern thoroughly.	For addressing the reviewer’s point, an alternative assay with measuring mean fluorescence intensity (MFI) for Figures 2e, 3e, 6f, and 6g (corresponding to new Fig 2f, Fig 3e , Fig, 7g and 7h) was performed to obtain more objective data. We found that the MFI of proinsulin in non-DM LSK BMT and DM LSK BMT shown in Figure 2e was 404.3 ± 20.54 and 479.5 ± 2.19 (p-value = 0.033), respectively. The MFIs of LSK-CD106(+) and Tnfα(+) in non-DM and STZDM shown in Fig. 3e were 204.5 ± 16.00 and 259.0 ± 10.3 (p-value = 0.025), respectively. Furthermore, the MFI of proinsulin shown in Figure 6f was 558.4 ± 19.4 and 689.7 ± 35.43, respectively (p value = 0.010), and the MFI of Tnf-α shown in Figure 6e was 194.2 ± 20.54 and 257.5 ± 17.62, respectively (p-value = 0.028). Although the rate of appearance of abnormal cells in LSK cells is small, we believe that the	

			appearance of 0.5-3% of abnormal stem cells is very significant because stem cells differentiate into peripheral blood throughout the body.	
2	Test whether the transplantation of CD106+ and CD106-LKS from non-diabetic mice have the same effect of the diabetic counterpart to confirm that CD106 identify putative LKS cells that triggers cell fusion and neuronal dysfunction.	It is important to get this experiment done.	This experiment was performed using diabetic and non-diabetic LSK cells based on the reviewer's comments. However, due to the low number of non-diabetic LSK/CD106 positive cells, we were unable to save the lives of the animals after irradiation. Therefore, we performed transplantation using CD106-positive or -negative mononuclear cells, and the results are shown in Figure 7. The results showed that some diabetic CD106-positive mononuclear cells are capable of cell fusion.	We have added these results (page 12) and new Figure 6 in the paper.
3	The authors should provide z-Stack images, with xyz projection of DRG to validate cell fusion. With the current images, it is difficult to assess whether cells are actually fused or just overlapping.	Please address this concern thoroughly.	Z-stack images with xyz projection of DRGs were performed to verify cell fusion in the models of Ayu1 TBM BMT Isltomato DM, DM LSK BMT, DM LSK CD106(+) BMT and 25% glucose ip.	These figures were shown in Figure 1f, 2e, 5e, 8e, respectively.
4	Explain the discrepancy of the proportion of CD106+ LKS in non-diabetic mice between figure 3d (1% of LKS cells positive for CD106) and figure 6h (5% of LKS cells positive for CD106).		In normal mice, the percentage of CD106 in LSK cells increased from 1% to 6% in response to intraperitoneal saline	

			injection. Likewise, as shown in new Fig. 8, the percentage of CD106 positive cells in LSK cells also increased from 5% to 12% in response to intraperitoneal injection of 25% glucose solution. These suggest that while intraperitoneal saline injection per se could slightly increase the number of CD106 in bone marrow, the stimulation of glucose solution was far potent compared to that of saline solution. The effect of glucose injection might be a combined effect of the stress from intraperitoneal injection and the effect of hyperglycemic stimulation.	
5	Characterize the regenerative capacity of ST-HSCs, which usually have a lower regenerative potential.		We verified the appearance of CD106-positive cells in LSK-CD34-positive and negative cells: the percentage of CD106 positive cells in the diabetic LSK-CD34 positive fraction as ST-HSCs was increased compared to non-diabetic LSK-CD34 positive cells. However, none of the CD106 positive cells appeared in the LSK-CD34 negative fraction. These data confirmed that diabetic abnormal CD106 positive cells were present only in	The following sentence has been put in the method section in Page 23. “we selected only mice with more than 90% of peripheral white blood cells of donor origin for this study”

			the ST-HSC fraction. In this study, the chimerism analysis of peripheral blood cells from all the bone marrow transplantation model mice showed that the chimerism of peripheral blood cells was more than 90%, suggesting that the reconstitution capacity of the bone marrow is normal.	
6	Did the authors have made any hypothesis about what kind of cells can fuse with DRG?	This is an interesting question. It will be great if you discuss this issue in the Discussion.	The model mice transplanted with nonDM CD106(-) MNCs and STZDM CD106(-) MNCs survived without weight loss or serious infections. These data indicate that even in the diabetic state, LT-HSCs without CD106-positive cells retain their stem cell integrity and have normal differentiation potential. Thus, hyperglycemia-induced abnormal ST-HSCs are common progenitor cells with abnormal differentiation potential for fusion with peripheral neurons, which are specific for diabetes and never appear in non-diabetes. We discussed above observation in discussion.	The following sentence has been put in the Discussion in Page 18. “It is reported that CD106 does not only express on inflammation-induced endothelial cells, but also participate in hemopoiesis, B cell development and T cell activation (47). These reports indicate that CD106 have also important role in immunological reaction. However, the model mice transplanted with nonDM CD106(-) MNCs and STZDM CD106(-) MNCs survived without weight loss or serious infections. These data indicate that even in the diabetic state, LT-HSCs without CD106-positive cells retain their stem cell integrity and have normal differentiation potential. Thus, hyperglycemia-induced abnormal ST-HSCs are common progenitor cells with abnormal differentiation potential for fusion with peripheral neurons, which are specific for diabetes and never appear in non-diabetes.”

REVIEWERS' COMMENTS:

Reviewer #1 (Remarks to the Author):

The authors have responded my questions

Reviewer #3 (Remarks to the Author):

The manuscript by Katagi and colleagues describes a mechanism by which hyperglycemia triggers the dysfunction of ST-HSCs that promote the onset of diabetic neuropathy by fusion of BM-derived cells with neurons.

The work is well written, and according to the rebuttal letter the authors replied convincingly and with the requested experiments.

However, I have a few comments that should be addressed.

-The authors reported that hyperglycemia but also transient hyperglycemia promotes a dysfunctional behavior of HSC. The authors should comment these findings in the discussion in the context of what is known about the effects of diabetes on the BM in terms of myelopoiesis and its altered functions (PMID: 30936144, 23663738).

-In line with these concepts, the authors insist that GFP+ cells fuse with neurons and trigger the onset of neuropathy. However, the authors did not define the identity of these cells apart from the expression of TNF-alpha and pro-insulin. It could be speculated that those cells are myeloid cells, which fit with the idea that diabetes rewire BM function by increasing myelopoiesis. The authors should comment about that.

-The authors showed that CD106 expression is a feature of diabetic ST-HSC. This is in line with previous finding that showed that hyperglycemia increased the expression of adhesion molecules in the diabetic BM (PMID: 24270983, 21998408). Please comment.

-While the experiment of transplanting CD106+ and CD106- LKS is well performed, I have some troubles in understanding the experiment with the antibody against CD106. It is not clear whether the antibody treatment reduces the engraftment of CD106+ cells, therefore favoring the engraftment of CD106- cells or if this experiment aimed to block the signaling of CD106 in driving the dysfunctional behavior of these cells. Alternatively if the authors aimed to deplete the CD106+ fraction, the authors should clarify. Importantly, provide the clone, the concentration used and the manufacturer of the CD106 monoclonal antibody.

Reviewer #1

The authors have responded my questions

Reviewer #3

The manuscript by Katagi and colleagues describes a mechanism by which hyperglycemia triggers the dysfunction of ST-HSCs that promote the onset of diabetic neuropathy by fusion of BM-derived cells with neurons. The work is well written, and according to the rebuttal letter the authors replied convincingly and with the requested experiments. However, I have a few comments that should be addressed.

-The authors reported that hyperglycemia but also transient hyperglycemia promotes a dysfunctional behavior of HSC. The authors should comment these findings in the discussion in the context of what is known about the effects of diabetes on the BM in terms of myelopoiesis and its altered functions (PMID: 30936144, 23663738).

【Response】

In contrast to the reviewer's understanding, we have reported that not only hyperglycemia but also transient hyperglycemia causes abnormal Proinsulin-producing cells to develop in various peripheral organs (Kojima H et al. PNAS 2004). In the present study, we have shown that transient hyperglycemia was potent enough to increase the number of abnormal CD106 cells expressing Tnf- α in the ST-HSC fraction and to cause neuropathy.

We now include the following sentence in the discussion;

"Recently, several researchers have shown interesting findings that diabetes could impair the mobilization of hematopoietic stem cells and myelopoiesis as a mechanism for diabetic complication⁴³.⁴⁴. The future study is warranted to investigate if HSC we found could be associated with abnormal myelopoiesis."

-In line with these concepts, the authors insist that GFP+ cells fuse with neurons and trigger the onset of neuropathy. However, the authors did not define the identity of these cells apart from the expression of TNF-alpha and pro-insulin. It could be speculated that those cells are myeloid cells, which fit with

the idea that diabetes rewire BM function by increasing myelopoiesis. The authors should comment about that.

【Response】

Since we found GFP+ cells expressing TNF-alpha and pro-insulin are bone marrow derived cells, we agree with the reviewer that these cells are myeloid cells.

-The authors showed that CD106 expression is a feature of diabetic ST-HSC. This is in line with previous finding that showed that hyperglycemia increased the expression of adhesion molecules in the diabetic BM (PMID: 24270983, 21998408). Please comment.

【Response】

Accordingly, we add the following sentence in the result section;

"Among these factors, we decided to focus on CD106 since this molecule is a marker of activated endothelial cells and activation of those cells is involved in the pathological process in diabetes (31, 32) and vascular formation (33). Previous studies also indicate the importance of adhesion molecule in hematopoietic stem/progenitor cells (HSPC)^{34, 35}"

-While the experiment of transplanting CD106+ and CD106- LKS is well performed, I have some troubles in understanding the experiment with the antibody against CD106. It is not clear whether the antibody treatment reduces the engraftment of CD106+ cells, therefore favoring the engraftment of CD106- cells or if this experiment aimed to block the signaling of CD106 in driving the dysfunctional behavior of these cells. Alternatively, if the authors aimed to deplete the CD106+ fraction, the authors should clarify. Importantly, provide the clone, the concentration used and the manufacturer of the CD106 monoclonal antibody.

【Response】

Accordingly, we now have included the following sentence in the methods section,

"Anti-CD106 antibody (biologend Inc., Clone 429 (MVCAM.A), final concentration:10µg/ml) was utilized in order to neutralize the adhesive action of CD106 mononuclear cells at DRG neurons".